# Identification of Risk Factors for Coronary Artery Disease in Asymptomatic Patients with Type 2 Diabetes Mellitus

**DOI:** 10.3390/jcm11051226

**Published:** 2022-02-24

**Authors:** Kazuhisa Takamura, Shinichiro Fujimoto, Tomoya Mita, Yuko Okano Kawaguchi, Mika Kurita, Satoshi Kadowaki, Yuki Kamo, Chihiro Aoshima, Yui Okada Nozaki, Daigo Takahashi, Ayako Kudo, Makoto Hiki, Nobuo Tomizawa, Fuki Ikeda, Hiroaki Satoh, Hirotaka Watada, Tohru Minamino

**Affiliations:** 1Department of Cardiovascular Biology and Medicine, Juntendo University Graduate School of Medicine, Tokyo 113-8421, Japan; k-takamu@juntendo.ac.jp (K.T.); yukawagu@juntendo.ac.jp (Y.O.K.); y-kawai@juntendo.ac.jp (Y.K.); caoshima@juntendo.ac.jp (C.A.); y-nozaki@juntendo.ac.jp (Y.O.N.); d-takahashi@juntendo.ac.jp (D.T.); a.kudo.gt@juntendo.ac.jp (A.K.); m-hiki@juntendo.ac.jp (M.H.); t.minamino@juntendo.ac.jp (T.M.); 2Department of Metabolism and Endocrinology, Juntendo University Graduate School of Medicine, Tokyo 113-8421, Japan; tom-m@juntendo.ac.jp (T.M.); mkurita@juntendo.ac.jp (M.K.); skadowa@juntendo.ac.jp (S.K.); fuki@juntendo.ac.jp (F.I.); hk-sato@juntendo.ac.jp (H.S.); hwatada@juntendo.ac.jp (H.W.); 3Department of Radiology, Juntendo University Graduate School of Medicine, Tokyo 113-8421, Japan; n-tomizawa@juntendo.ac.jp; 4Japan Agency for Medical Research and Development-Core Research for Evolutionary Medical Science and Technology (AMED-CREST), Japan Agency for Medical Research and Development, Tokyo 100-0004, Japan

**Keywords:** coronary artery disease, asymptomatic, type 2 diabetes mellitus, non-invasive examination

## Abstract

Background: Patients with diabetes mellitus (DM) are a high-risk group for coronary artery disease (CAD). In the present study, we investigated predictive factors to identify patients at high risk of CAD among asymptomatic patients with type 2 DM based on coronary computed tomographic angiography (CCTA) findings. Methods: A single-center prospective study was performed on 452 consecutive patients with type 2 DM who were provided with a weekly hospital-based diabetes education program between 3 October 2015, and 31 March 2020. A total of 161 consecutive asymptomatic patients (male/female: 111/50, age: 57.3 ± 9.3 years) with type 2 DM without any known CAD underwent CCTA. Based on conventional coronary risk factors and non-invasive examination, i.e., measurement of intima-media thickness, subcutaneous and visceral fat area, a stress electrocardiogram test, and the Agatston score, patients with obstructive CAD, CT-verified high-risk plaques (CT-HRP), and optimal revascularization within 90 days were evaluated. Results: Current smoking (OR, 4.069; 95% C.I., 1.578–10.493, *p* = 0.0037) and the Agatston score ≥100 (OR, 18.034; 95% C.I., 6.337–51.324, *p* = 0.0001) were independent predictive factors for obstructive CAD, while current smoking (OR, 5.013; 95% C.I., 1.683–14.931, *p* = 0.0038) was an independent predictive factor for CT-HRP. Furthermore, insulin treatment (OR, 5.677; 95% C.I., 1.223–26.349, *p* = 0.0266) was the only predictive factor that correlated with optimal revascularization within 90 days. Conclusions: In asymptomatic patients with type 2 DM, current smoking, an Agatston score ≥100, and insulin treatment were independent predictive factors of patients being at high-risk for CAD. However, non-invasive examinations except for Agatston score were not independent predictors of patients being at high risk of CAD.

## 1. Introduction

Patients with diabetes mellitus (DM) are a high-risk group for coronary artery disease (CAD) because they more frequently develop CAD than non-DM patients, and CAD progresses without symptoms and occasionally follows a serious clinical course [1,2,3,4,5]. Although myocardial ischemia is detected in approximately 22% of asymptomatic patients with DM, many of these patients are low-risk patients without myocardial ischemia [6], and, thus, are a heterogenous group comprising patients with and without CAD. No screening methods for asymptomatic patients with DM using a non-invasive examination for CAD other than the Agatston score are currently recommended by guidelines [7,8,9], and the Agatston score is not covered by the national health insurance system in Japan. However, it is important to efficiently identify high-risk groups in asymptomatic patients with DM for early therapeutic interventions and the prevention of cardiovascular events.

Among asymptomatic patients with type 2 DM receiving various antidiabetic medications that are expected to effectively prevent atherosclerosis-related disease [10,11], predictive factors were examined in high-risk patients with obstructive CAD and computed tomography-verified high-risk plaque (CT-HRP) [12,13,14,15,16] and those who underwent revascularization within 90 days based on coronary CT angiography (CCTA) findings as a reference.

## 2. Materials and Methods

### 2.1. Study Population

A single-center prospective study was performed on 452 consecutive patients with type 2 DM aged 35–70 years who were provided with a weekly hospital-based diabetes education program between 3 October 2015 and 31 March 2020 and underwent blood tests, measurement of intima-media thickness (IMT), subcutaneous fat area (SFA), and visceral fat area (VFA), and a stress electrocardiogram (ECG) test as screening examinations. After the exclusion of patients with known CAD (angina pectoris and old myocardial infarction) (N = 97), those with intolerance to CCTA (bronchial asthma (N = 15), severe aortic valve stenosis (N = 4), allergy to contrast medium (N = 5), and chronic kidney disease (N = 62)), and those who did not consent (N = 70), CCTA was performed using 320-row area-detector CT (ADCT), and a total of 161 consecutive asymptomatic patients (male/female: 111/50; age: 57.3 ± 9.3 years old) with type 2 DM were evaluated by excluding those with low-quality images (motion artifact (N = 22), pacemaker leads (N = 11), poor contrast medium (N = 3), and a lack of data (N = 2)) (Figure 1).

#### 2.1.1. CCTA Acquisition Methods

The CT system used was Aquilion ONE ViSION Edition™ or Aquilion ONE GENESIS Edition™ (320-ADCT, Canon Medical Systems Corporation, Otawara, Japan). The automatic contrast medium injector was Dual Shot (Nemoto Kyorindo Co., Ltd., Tokyo, Japan), the image analyzer was ZIOSTATION (M900, Ziosoft Inc., Tokyo, Japan), and the ECG monitor was Model 7800 (Chronos Medical Device Inc., Tokyo, Japan). As a pretreatment, metoprolol was orally administered at 20–40 mg on the day of the examination at a heart rate (HR) ≥61 bpm unless there was a contraindication (systolic blood pressure < 90 mmHg, severe atrioventricular block, heart failure). If the HR remained at ≥61 bpm even after the administration of metoprolol, it was controlled by an intravenous injection of landiolol at 12.5 mg.

#### 2.1.2. CCTA Protocol

Contrast medium (Omnipaque 350 mg/mL; Daiichi Sankyo Company, Tokyo, Japan) was injected via the cubital vein using a two-step method at a rate of body weight × 0.06 mL/s over a fixed time of 12 s, followed by an injection of 30 mL of physiological saline at the same rate.

Imaging conditions were a scanning slice thickness of 0.5 mm × 320 rows, an image slice thickness of 0.5 mm, and a reconstruction interval of 0.25 mm. The minimum number of rows needed to cover all coronary arteries was selected from 200 rows (100 mm), 240 rows (120 mm), 256 rows (128 mm), 280 rows (140 mm), and 320 rows (160 mm), with reference to unenhanced CT images obtained to calculate the coronary artery calcium score (CACS).

The tube voltage was 100 kV (body mass index: BMI > 30, 120 kV), and the tube current was the mean tube current level calculated using the automatic exposure control (AEC) function with a standard deviation (SD) of 20. Scanning was performed by a single rotation, at a gantry rotation speed of 275 ms/rot., in the prospective CTA mode, and by setting the X-ray exposure to a range of 70–99% of the RR interval.

Adaptive iterative dose reduction using three-dimensional processing (AIDR3D: Toshiba, Medical Systems) was used for all patients with an intensity at the standard setting.

#### 2.1.3. Measurement of the Agatston Score

Using position-setting images before the implementation of CCTA, CACS was assessed using the method of Agatston et al. [17]. Imaging was performed in an area covering the aortic root and cardiac apex by simple cardiac CT targeting mid-diastole or end-systole with prospective ECG gating using 280 rows of detectors at a tube voltage of 120 kV and a tube current of 50 mA. The obtained images were reconstructed at a slice thickness of 3 mm and a slice interval of 3 mm. A calcified lesion was defined as ≥3 contiguous pixels with a peak attenuation ≥130 Hounsfield unit (HU) using software on ZIOSTATION (M900, Ziosoft Inc., Tokyo, Japan). Total CACS was calculated according to the method of Agatston et al. [17].

#### 2.1.4. Evaluation of the Degree of Stenosis and Classification of CAD

CCTA findings were evaluated in each segment according to the modified American heart association classification based on agreement among 2 cardiologists and one radiologist without clinical information on patients [18]. The degree of stenosis was assessed in the axial, curved-multiplanar reconstruction angiographic view. The percentage ratio of the stenotic lumen to the original vessel diameter of the lesion, analogized by a presumed-to-be-healthy distal site and proximal to the site of stenosis, was obtained, and the degree of stenosis was expressed by subtracting this from 100.

Obstructive CAD was defined as lesions with >50% diameter stenosis on CCTA in ≥1 vessel or those in which stenosis was difficult to evaluate due to severe calcification. The lesion index was classified as normal when no plaque was observed in the coronary artery, 0 vessels when plaques were observed but the degree of stenosis was <50%, 1 vessel when the degree of stenosis was ≥50%, and 2 vessels when there were plaques in the left main trunk (LMT). Vessels ≥2.0 mm in diameter were examined. In addition, the non-obstructive CAD group was defined as a group other than obstructive CAD.

#### 2.1.5. Definition of CT-HRP

CT-HRP was defined as plaque accompanied by 2 or more of the following: positive remodeling (PR), low attenuation plaque (LAP), spotty calcification, and the napkin-ring sign [12].

PR was defined as a change in the coronary artery diameter at the plaque site relative to the reference segment with a normal appearance (reference diameter). The remodeling index was defined as the lesion diameter divided by the reference diameter, and measurements were made using cross-sectional and longitudinal reconstructions. The presence of PR was defined when the coronary diameter at the plaque site was at least 10% larger than the reference segment. Attenuation was defined as the minimum HU among five 0.36 × 0.36-mm regions of interest. The lesion was defined as LAP when the minimum HU was <30 [13,14] Spotty calcification was defined as being <3 mm in size on curved multiplanar reformation images and one side on cross-sectional image [13]. The napkin-ring sign was defined as the presence of a ring of higher HU value than the adjacent plaque and lower than 130 HU [15,16].

### 2.2. Measurement of Carotid IMT

The carotid IMT was evaluated using LOGIQGE P6 (GE Healthcare Systems, Chicago, IL, USA). Measurements were made in observable areas of the proximal wall, distal wall, and walls on both sides of the bilateral common carotid arteries (CCA). Carotid IMT was defined as the distance between the high-echo layer on the intimal side (luminal side of the vessel) (leading edge: LE) and the outer high-echo layer in the long-axis view of the carotid artery.

The mean values of three points every 10 mm from the bifurcation of bilateral CCA were calculated and defined as Lt mean IMT and Rt mean IMT, respectively. The mean IMT was defined as (Lt mean IMT + Rt mean IMT)/2. The max IMT was defined as the site of greatest thickness, including a plaque lesion, in either CCA. Furthermore, increased IMT was defined as IMT to ≥1.1 mm at the measurement site in either CCA [19].

### 2.3. Measurement of the Subcutaneous Fat Area (SFA) and Visceral Fat Area (VFA)

The total fat area and VFA of HU −190 to −30 were measured at a cross-section of the navel vertebra. SFA was calculated as the difference between the total fat area and VFA. The visceral fat-type obesity was defined as VFA/SFA > 0.4 [20].

### 2.4. Evaluation of Stress ECG Test

The stress ECG test was performed using Master’s double two-step method [21]. The obtained results were positive when a decrease of ≥0.1 mV was observed from the ST segment 0.08 s after the J point, and the ST segment was horizontal or descending.

### 2.5. Definition of Risk Factors and Evaluation of Diabetic Complications

Patients with a fasting blood glucose level of ≥126 mg/dL, a casual blood glucose level of ≥200 mg/dL, or a HgbA1c of ≥6.5% according to the National Glycohemoglobin Standardization Program were defined as the diabetic type. In general, for the diagnosis of diabetes, either of the following criteria is to be followed: 1. Two assessments of the diabetic type in each patient (where one blood glucose test is mandatory). 2. One assessment of the diabetic type (with mandatory blood glucose testing) along with the presence of typical symptoms of chronic hyperglycemia (e.g., dry mouth, polydipsia, polyuria, body weight loss, or diabetic retinopathy). 3. Evidence of a prior diagnosis of diabetes [22].

Patients with a blood pressure of ≥140/90 mmHg or those already being administered antihypertensive drugs were defined as hypertensive. Patients with a total cholesterol (T-Cho) level of ≥220 mg/dL, a low-density lipoprotein-cholesterol (LDL-C) level of ≥140 mg/dL, a fasting triglyceride level of ≥150 mm/dL, or a high-density lipoprotein-cholesterol level of ≤40 mg/dL, and those already being administered antihyperlipidemic drugs were defined as having dyslipidemia. Patients who had smoked within 1 year before the CT examination were defined as current smokers. Among diabetic complications, simple retinopathy or more advanced retinopathy according to the Davis classification was defined as diabetic retinopathy [23]. Nephropathy with a urinary albumin/creatinine ratio of ≥30 mg/g • Cr was defined as diabetic nephropathy [24]. A condition that fulfilled 2 or more of the following criteria, (1) positive findings in the bilateral toes/feet and lower legs (tingling numbness, stinging, cutting, burning, or aching pain), (2) symmetric hypesthesia of distal parts of the lower legs, and (3) bilateral attenuation or the absence of the Achilles tendon reflex, was defined as diabetic neuropathy according to the diagnostic criteria of the Toronto Diabetic Neuropathy Expert Group [25].

### 2.6. Optimal Revascularization within 90 Days

The Optimal revascularization was defined as (1) ≥90% stenotic lesion, (2) ≥50% stenotic lesion of the LMT and left-anterior descending branch, (3) 2-vessel or 3-vessel lesion with an ejection fraction of ≤40%. (In case (2) and (3), the presence of ischemia in ≥10% of the left ventricle was demonstrated by myocardial perfusion imaging or a fraction flow reserve of ≤0.8.) [26].

### 2.7. Statistical Analysis

Statistical analyses were performed using Statview J-5.0 for Windows (HULINKS, Inc., Tokyo, Japan) and MedCalc Version 12.2.1. (MedCalc Software bvba, Mariakerke, Belgium).

Continuous variables were expressed as means ± SD or the median (minimum–maximum) and categorical variables as percentages.

The Mann–Whitney U test and χ^2^ test were used for the comparison of clinical characteristics including non-invasive examination findings between the non-obstructive group and the obstructive CAD group. Predictive factors for CT-HRP and those who underwent optimal revascularization within 90 days were evaluated by a univariate analysis using a logistic regression analysis of age, gender, duration of diabetes, BMI, conventional coronary artery risk factors, inflammation markers, diabetic complications (diabetic ophthalmopathy, nephropathy, and neuropathy), oral medications, and the results of non-invasive examination, i.e., mean IMT, max IMT, increased IMT (IMT ≥ 1.1 mm), visceral fat-type obesity (VFA/SFA > 0.4), stress ECG test results, the Agatston score, and the Agatston score category (0–99,100–).

Moreover, in patients with obstructive CAD or CT-HRP and those who underwent optimal revascularization within 90 days, independent predictive factors were identified by a multivariate analysis of factors that were significant in the comparison between 2 groups and the univariate analysis using a logistic regression analysis. The level of significance was set at *p* < 0.05.

## 3. Results

### 3.1. Patient Characteristics

Patient characteristics are shown in Table 1. The duration of diabetes was 10.7 ± 8.3 years, and that of HbA1c was 8.6 ± 1.5% years. Statins were administered to 49.1% (79/161) of patients.

Non-invasive examinations revealed the visceral fat-type obesity in 81.4% (131/161) of patients and increased IMT in 31.1% (50/161). The stress ECG test was positive in 11.2% of patients (18/161). Obstructive CAD was observed in 26.7% (43/161) of patients and CT-HRP in 16.8% (27/161). Furthermore, 7.5% of patients (12/161) underwent optimal revascularization within 90 days, and none underwent revascularization without indications (Table 1).

### 3.2. Evaluation of Predictive Factors for Obstructive CAD

Comparing the non-obstructive CAD and obstructive CAD groups, current smoking (21.2 vs. 44.2%, *p* < 0.0038) and diabetic neuropathy (15.3 vs. 34.9%, *p* < 0.0063) were significant factors of obstructive CAD. According to the results of non-invasive examinations, max IMT (1.05 ± 0.55 vs. 1.36 ± 0.78 mm, *p* < 0.0059), increased IMT (26.3 vs. 44.2, *p* < 0.0297), the Agatston score (291.4 (582.8–0) vs. 135.3 (1586.3–12.8), *p* < 0.0001), and the Agatston score category (0–99/100–) (93.2/6.8 vs. 44.2/55.8%, *p* < 0.0001)) were significant factors for obstructive CAD.

Age, the duration of diabetes, conventional coronary risk factors, including high-sensitivity C-reactive protein, or oral medication for risk management were not predictive factors. Moreover, the visceral fat-type obesity and a positive stress ECG test were not significant factors (Table 2).

When a multivariate analysis was performed on current smoking, diabetic neuropathy, max IMT, and increased IMT, which were identified as significant factors in the comparison between the non-obstructive CAD and obstructive CAD group, by excluding an Agatston score ≥100, which is effective for the risk stratification of obstructive CAD, current smoking (OR, 2.902; 95% C.I., 1.317–6.396, *p* < 0.0082) and diabetic neuropathy (OR, 2.882; 95% C.I., 1.221–6.803, *p* < 0.0157) were identified as independent predictive factors (Table 3).

In the multivariate analysis with the addition of an Agatston score ≥100, only current smoking (OR, 4.069; 95% C.I., 1.578–10.493, *p* < 0.0037) and an Agatston score ≥100 (OR, 18.034; 95% C.I., 6.337–51.324, *p* < 0.0001) were identified as independent predictive factors for obstructive CAD (Table 3).

### 3.3. Evaluation of Predictive Factors for CT-HRP

When a univariate analysis using a logistic regression analysis was performed for predictive factors for CT-HRP, gender (male) (OR, 4.322; 95% C.I., 1.236–15.109, *p* < 0.0219), current smoking (OR, 4.333; 95% C.I., 1.832–10.250, *p* < 0.0008), T-Cho (OR, 1.013; 95% C.I., 1.001–1.024, *p* < 0.0293), TG (OR, 1.009; 95% C.I., 1.004–1.014, *p* < 0.0008), non-HDL (OR, 1.017; 95% C.I., 1.005–1.028, *p* < 0.0058), fasting glucose (OR, 1.017; 95% C.I., 1.005–1.028, *p* < 0.0037), glycated albumin (OR, 1.077; 95% C.I., 1.004–1.157, *p* < 0.0398), and diabetic neuropathy (OR, 2.829; 95% C.I., 1.153–6.989, *p* < 0.0232) were identified as predictive factors (Table 4), while indices of non-invasive examinations, such as an Agatston score ≥100, IMT parameter, visceral fat-type obesity, and positive stress ECG test results were not.

When a multivariate analysis was performed on factors identified by a univariate analysis as predictive factors for CT-HRP, only current smoking (OR, 5.013; 95% C.I., 1.683–14.931, *p* < 0.0038) was an independent predictive factor for CT-HRP (Table 5).

### 3.4. Evaluation of Predictive Factors for Optimal Revascularization within 90 Days

In a univariate logistic regression analysis, T-Cho (OR, 1.017; 95% C.I., 1.001–1.033, *p* < 0.0332), LDL-C (OR, 1.027; 95% C.I., 1.007–1.048, *p* < 0.0077), non-HDL (OR, 1.018; 95% C.I., 1.002–1.034, *p* < 0.0263), diabetic retinopathy (OR, 3.913; 95% C.I., 1.143–13.397, *p* < 0.0298), and insulin treatment (OR, 6.313; 95% C.I., 1.635–24.377, *p* < 0.0075) were identified as predictive factors. Agatston score ≥100, IMT parameter, visceral fat-type obesity, and positive stress ECG test results were not shown to be predictive factors (Table 6).

When a multivariate analysis was performed on factors found by multivariate analysis to be predictive factors, only insulin treatment (OR, 5.677; 95% C.I., 1.223–26.349, *p* < 0.0266) was identified as a predictive factor (Table 7).

## 4. Discussion

Although patients with DM are widely recognized as a high-risk group for CAD [1,2,3,4,5], screening using non-invasive examinations did not contribute to improvements in patient outcomes in any large-scale clinical studies, including the DIAD study [6], which used myocardial perfusion imaging on asymptomatic patients with DM, and the FACTOR-64 study [27], which used CCTA. Therefore, the screening of asymptomatic patients with DM for CAD using non-invasive examinations is not currently recommended. Only 22% of patients in the DIAD study [6] had myocardial ischemia, while 83% of those in the FACTOR-64 study [27] had no significant stenosis in the coronary artery. In the present study, 73.3% (118/161) of patients had no significant stenosis in the coronary artery, 83.2% (134/161) had no CT-HRP, and 92.5% (149/161) did not undergo optimal revascularization within 90 days, which supports the view that asymptomatic patients with type 2 DM are a heterogeneous group comprising patients with and without CAD.

To efficiently identify patients at high risk of CAD in the heterogeneous group of asymptomatic patients with type 2 DM, we herein investigated predictive factors for obstructive CAD, CT-HRP, and optimal revascularization within 90 days, which are high-risk factors for CAD, based on CCTA findings as a reference.

The Agatston score is currently recommended by guidelines as a screening method for obstructive CAD in high-risk asymptomatic patients [7,8,9]. However, since it is not yet covered by the national health insurance system in Japan, we also examined predictive factors other than the Agatston score. The results obtained showed that current smoking and diabetic neuropathy were the significant factors for obstructive CAD, and, among non-invasive examination results, visceral fat-type obesity, including IMT, and a positive stress ECG test were not significant factors. In patients with DM, impaired vascular intimal function has been reported to progress from an early stage due to protein glycosylation and increased oxidative stress [28,29], and smoking, which is closely involved in the initiation, maintenance, and progression of atherosclerosis, clearly promotes its progression to obstructive CAD [30,31]. The present results are also reasonable in consideration of the view that smoking itself is an independent risk factor for the development of cardiovascular events and death [32,33]. Furthermore, diabetic neuropathy was also a predictor of obstructive CAD, and the resulting autonomic neuropathy and denervation may affect the development of silent myocardial ischemia and may be one mechanism by which CAD becomes more severe while patients with DM are asymptomatic [34,35].

In contrast, in a multivariate analysis using a logistic analysis with the addition of the Agatston score ≥100 as another non-invasive index for these factors, only current smoking and an Agatston score ≥100 were identified as independent predictive factors for obstructive CAD. The Agatston score was previously shown to be useful for the risk stratification of cardiovascular events in asymptomatic patients [36,37], and cardiovascular events were reported to occur in 1.5%/year of patients with an Agatston score ≥100 and in 2.4%/year of patients with an Agatston score ≥400 [38]. One of the reasons for this is that the Agatston score, which is based on a direct measurement of coronary artery calcification, represents the extent and severity of atherosclerosis [39,40], and the results support the guidelines [7,8,9].

According to our results on CT-HRP, only current smoking was an independent predictive factor for obstructive CAD. Smoking is cytotoxic because it increases radical oxygen production and nitric oxide (NO) binding. Therefore, smoking directly damages vascular wall cells, reduces the biological activity of NO in vascular endothelial cells and vascular smooth muscle cells, and results in oxidative stress, which has been implicated in plaque instability as a result of the impairment of vascular endothelial function and the causing of inflammation [41,42]. However, in contrast to obstructive CAD, the Agatston score was not identified as a significant predictive factor for CT-HRP. This result is interesting because CT-HRP is generally considered to be a lesion that is rich in fat with no or only slight calcification, typically spotty calcification [13,14,15,16]; therefore, the Agatston score was not a predictive factor for CT-HRP.

Regarding optimal revascularization within 90 days, insulin treatment was the only predictive factor. The ORIGIN study in patients with impaired glucose tolerance and early type 2 DM showed that treatment with the insulin-glargine did not increase composite cardiovascular outcomes compared to standard care without insulin [43]. On the other hand, the use of insulin to reduce blood glucose levels is limited to patients with advanced diabetes, and no regimens have been reported to improve patient outcomes [44]. Furthermore, it was reported that iatrogenic hyperinsulinemia promotes an inflammatory macrophage response [45]. In the present study, the duration of diabetes was 10.7 ± 8.3 years, and these results may have been influenced by long-term inappropriate insulin treatment that failed to adequately regulate blood glucose. In addition, since the selection of revascularization currently requires the confirmation of not only a high degree of stenosis, but also ischemia [26], the Agatston score, the IMT parameter measured by carotid artery ultrasonography, and visceral fat-type obesity may be excluded as independent predictive factors. Furthermore, while a stress ECG test is a convenient examination for the evaluation of ischemia, its sensitivity is limited, which is also considered to explain its elimination [46].

In the present study, non-invasive examinations that are widely used to screen for atherosclerosis, such as the measurement of IMT by carotid artery ultrasonography, the measurement of SFA/VFA by abdominal CT, and the stress ECG test, were not independent predictive factors for obstructive CAD in advanced-stage atherosclerosis, CT-HRP suggestive of unstable plaques, or optimal revascularization within 90 days. Previous studies reported that increased IMT did not correlate with an increase in cardiovascular events in patients with DM [47]. Furthermore, the European Society of Cardiology (ESC) guidelines do not recommend these examinations as screening tests and rate them as IIIb [8], and the present study did not reverse these proposals. Moreover, the judgment of visceral fat-type obesity based on the VFA/SFA by abdominal CT is not recommended in Western countries because it is not an index of metabolic abnormalities in severe obesity [48]. Furthermore, the Japanese guidelines recommend stress ECG as the first test for CAD risk stratification [7]; however, its role as a screening test for the risk of CAD in asymptomatic patients with DM is minimal in consideration of its elimination as a predictive factor in the present study.

The ESC guidelines state that CCTA is an effective examination, but do not recommend it as a screening test for asymptomatic patients due to the lack of evidence to support improvements in patient outcomes. However, the SCOT-HEART trial reported that mortality due to CAD and the incidence of non-lethal myocardial infarction were significantly lower when CCTA was added to standard treatments for patients with stable angina pectoris than with standard treatments alone [49].

Since none of the non-invasive screening examinations were independent predictors of patients being at high risk of CAD in the present study, it may be possible to efficiently select high-risk groups for CAD among asymptomatic patients with type 2 DM by selectively performing CCTA on smokers and insulin users, who are at a high risk of developing CAD, and this will contribute to risk stratification and improvements in patient outcomes through early multifactorial therapeutic interventions, as previous reported [50], as well as the prevention of cardiovascular events.

## 5. Study Limitations

There are several limitations in the present study. First, although the present study was an observational study, CAD findings by CCTA during the hospitalization period were used as the outcome, which makes it difficult to interpret the causal relationship between CAD and risk factors. Second, it was a single-center study, included a relatively low number of subjects, and there may have been bias in the study population because it was not possible to obtain consent from some patients. In addition, low prevalence may have reduced the power of the predictor. Therefore, a multi-center study is needed in the future. Second, we also examined predictive factors based on CCTA findings. Patients with severe calcification were also included, and since this may have not only led to the overestimation of coronary artery stenosis, but may also have affected the evaluation of plaque properties, there may have been differences from the true prevalence or plaque properties. Fourth, albuminuria is an important risk factor for CAD in symptomatic patients with type 2 DM [51].

Finally, the duration of oral medication and insulin therapy was unknown. Since the time of the initiation of antidiabetic or lipid-lowering drug treatment was selected according to patient characteristics and by the judgment of the physician in charge, the effects of these drugs on atherosclerosis were not evaluated in the present study.

## 6. Conclusions

Asymptomatic patients with type 2 DM were a heterogeneous group that included those without CAD. An Agatston score ≥100 and current smoking were independent predictive factors for obstructive CAD, while current smoking was a predictive factor for CT-HRP. The insulin treatment was the only predictive factor for optimal revascularization within 90 days. However, non-invasive examinations other than assessment of the Agatston score were not independent predictors of patients being at high risk of CAD.

## Figures and Tables

**Figure 1 jcm-11-01226-f001:**
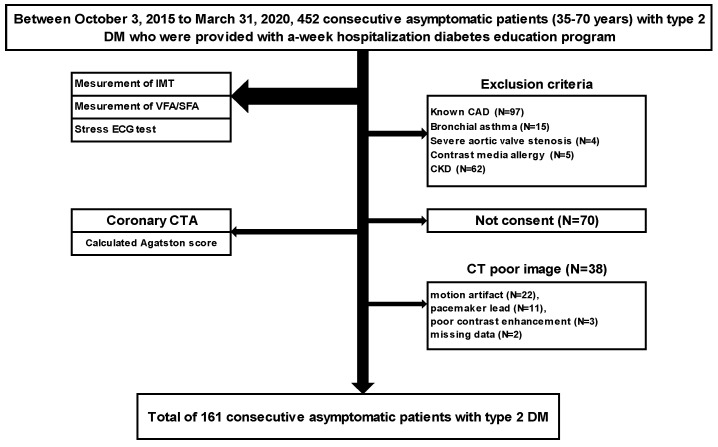
Study Protocol. DM, diabetes mellitus; IMT, intima-media thickness; VFA, visceral fat area; SFA, subcutaneous fat area; ECG, electrocardiogram; CTA, computed tomography angiography; CAD, coronary artery disease; CKD, chronic kidney disease.

**Table 1 jcm-11-01226-t001:** Patient characteristics.

	All (N = 161)		All (N = 161)
Gender (Men %), (n)	68.9 (111)	Retinopathy (%), (n)	17.4 (28)
Age (years old)	57.3 ± 9.3	Nephropathy (%), (n)	23.6 (38)
Duration of diabetes (years)	10.7 ± 8.3	Neuropathy (%), (n)	20.5 (33)
BMI (kg/m^2^)	26.4 ± 4.4	Medical treatment	
HT (%), (n)	37.9 (61)		
DL (%), (n)	66.5 (107)	Insulin treatment (%), (n)	35.4 (57)
Current smoking (%), (n)	28.0 (45)	Metformin (%), (n)	44.7 (72)
T-Cho (md/dL)	193.8 ± 36.7	Pioglitazone (%), (n)	7.5 (12)
TG (md/dL)	151.9 ± 36.6	DPP-4I (%), (n)	40.4 (65)
LDL-C (md/dL)	118.9 ± 31.8	SGLT2I (%), (n)	23.6 (38)
HDL-C (md/dL)	48.6 ± 12.4	ACE-I/ARB (%), (n)	29.2 (47)
non-HDL (md/dL)	148.4 ± 35.9	Statin (%), (n)	49.1 (79)
Lp(a) (mg/dL)	15.1 ± 15.6	Non-invasitve examination	
UA (mg/dL)	5.5 ± 1.2		
HbA1c (%)	8.6 ± 1.5	Agatston score	10.5 (0.0–4295.7)
Fasting glucose (g/dL)	145.0 ± 49.8	Agatston score category	80.1 (129)/19.9 (32)
GA (mg/dL)	20.2 ± 6.1	0–99/100– (%), (n)	
hs-CRP (mg/dL)	0.2 ± 0.4	Obstructive CAD (%), (n)	26.7 (43)
Cre (mg/dL)	0.74 ± 0.72	Vessel disease N/0/1/2/3 (n)	44/74/29/11/3
eGFR (mL/min/1.73 m^2^)	88.5 ± 18.7	CT-HRP (%), (n)	16.8 (27)
		Visceral fat-type obesity (%), (n)	81.4 (131)
		Mean IMT (mm)	0.80 ± 0.24
		Max IMT (mm)	1.13 ± 0.63
		Increased IMT (%), (n)	31.1 (50)
		Stress ECG positive (%), (n)	11.2 (18)
		Optimal revascularization within 90 days (%), (n)	7.5 (12)

BMI, body mass index; HT, hypertension; DL, dyslipidemia; T-Cho, total cholesterol; TG, triglyceride; LDL-C, low-density lipoprotein cholesterol; HDL-C, high-density lipoprotein cholesterol; non-HDL, non-high-density lipoprotein; Lp, lipoprotein; UA, uric acid; HbA1c, hemoglobin A1c; GA, glycated albumin; hs-CRP, high-sensitivity C-reactive protein; Cre, creatinine; eGFR, estimated glomerular filtration rate; DPP-4I, dipeptidyl peptidase-4 inhibitor; SGLT2I, sodium glucose cotransporter-2 inhibitor; ACE-I, angiotensin-converting enzyme inhibitor; ARB, angiotensin II receptor blocker; CAD, coronary artery disease; CT-HRP, computed tomography-verified high-risk plaque; IMT, intima-media thickness; ECG, electrocardiogram.

**Table 2 jcm-11-01226-t002:** Comparison of clinical characteristics including non-invasive examination findings between non-obstructive group and obstructive CAD group.

	Non-Obstructive CAD (N = 118)	Obstructive CAD (N = 43)	*p*-Value		Non-Obstructive CAD (N = 118)	Obstructive CAD (N = 43)	*p*-Value
Gender (Men %)	65.3	79.1	0.0937	Retinopathy (%)	14.4	25.6	0.0979
Age (years old)	56.5 ± 9.4	59.7 ± 8.8	0.0549	Nephropathy (%)	24.6	20.9	0.6298
Duration of diabetes(years)	9.9 ± 7.8	12.8 ± 9.4	0.0508	Neuropathy (%)	15.3	34.9	0.0063
BMI (kg/m^2^)	26.7 ± 4.5	25.6 ± 4.0	0.1682	Medical treatment			
HT (%)	33.9	48.8	0.0838				
DL (%)	66.1	67.4	0.8734	Insulin treatment (%)	36.4	32.6	0.6486
Current smoking (%)	21.2	44.2	0.0038	Metformin (%)	44.9	44.2	0.9344
T-Cho (md/dL)	193.8 ± 36.7	205.7 ± 35.4	0.0684	Pioglitazone (%)	6.8	9.3	0.5897
TG (md/dL)	145.5 ± 76.4	169.4 ± 149.7	0.1863	DPP-4I (%)	39.0	44.2	0.5516
LDL-C (md/dL)	116.5 ± 31.6	125.5 ± 31.8	0.1147	SGLT2I (%)	25.4	18.6	0.3673
HDL-C (md/dL)	48.6 ± 12.9	48.5 ± 11.3	0.9616	ACE-I/ARB (%)	25.4	39.5	0.0814
non-HDL (md/dL)	145.2 ± 34.9	157.1 ± 37.3	0.0605	Statin (%)	48.3	51.2	0.7483
Lp(a) (mg/dL)	15.1 ± 15.8	14.9 ± 15.0	0.9343	Non-invasive examination			
UA (mg/dL)	5.4 ± 1.2	5.7 ± 1.3	0.1447				
HbA1c (%)	8.6 ± 1.5	8.5 ± 1.7	0.7380	Agatston score	291.4 (582.8–0)	135.3 (1586.8–12.8)	0.0001
Fasting glucose (g/dL)	142.2 ± 37.1	152.6 ± 74.3	0.2453	Agatston score category	93.2/6.8	44.2/55.8	0.0001
GA (mg/dL)	20.1 ± 5.9	20.6 ± 6.7	0.6093	0–99/100– (%)			
hs-CRP (mg/dL)	0.2 ± 0.5	0.1 ± 0.2	0.9343	Visceral fat-type obesity (%)	78.0	90.7	0.0664
Cre (mg/dL)	0.7 ± 0.8	0.7 ± 0.2	0.8246	Mean IMT (mm)	0.78 ± 0.25	0.82 ± 0.18	0.3534
eGFR (mL/min/1.73 m^2^)	89.7 ± 18.6	85.2 ± 18.7	0.1783	Max IMT (mm)	1.05 ± 0.55	1.36 ± 0.78	0.0059
				Increased IMT (%)	26.3	44.2	0.0297
				Stress ECG positive (%)	9.3	16.3	0.2152

CAD, coronary artery disease; 95% C.I., 95% confidence interval; BMI, body mass index; HT, hypertension; DL, dyslipidemia; T-Cho, total cholesterol; TG, triglyceride; LDL-C, low-density lipoprotein cholesterol; HDL-C, high-density lipoprotein cholesterol; non-HDL, non-high-density lipoprotein; LP, lipoprotein; UA, uric acid; HbA1c, hemoglobin A1c; GA, glycated albumin; hs-CRP, high-sensitivity C-reactive protein; Cre, creatinine; eGFR, estimated glomerular filtration rate; DPP-4I, dipeptidyl peptidase-4 inhibitor; SGLT2I, sodium glucose cotransporter-2 inhibitor; ACE-I, angiotensin-converting enzyme inhibitor; ARB, angiotensin II receptor blocker; IMT, intima-media thickness; ECG, electrocardiogram.

**Table 3 jcm-11-01226-t003:** Adjusted odds ratio of obstructive coronary artery disease in coronary computed tomographic angiography for clinical and carotid intima media thickness.

			Analysis in Addition to Agatston Score		
	Obstructive CAD	*p*-Value		Obstructive CAD	*p*-Value
	Odds Ratio (95% C.I.)			Odds Ratio (95% C.I.)	
Current smoking (%)	2.902 (1.317–6.396)	0.0082	Current smoking (%)	4.069 (1.578–10.493)	0.0037
Neuropathy (%)	2.882 (1.221–6.803)	0.0157	Neuropathy (%)	1.711 (0.618–4.734)	0.3010
Non-invasive examination			Non-invasive examination		
Max IMT (mm)	1.687 (0.857–3.320)	0.1301	Agatston score category		
Increased IMT (%)	1.610 (0.616–4.208)	0.3316	0–99 (reference)	1 (1)	N.A.
			100– (%)	18.034 (6.337–51.324)	0.0001
			Max IMT (mm)	1.530 (0.684–3.423)	0.3004
			Increased IMT (%)	1.698 (0.560–5.145)	0.3493

CAD, coronary artery disease; 95% C.I., 95% confidence interval; IMT, intima-media thickness.

**Table 4 jcm-11-01226-t004:** Univariate analysis using logistic regression to identify the independent predictor of CT-HRP.

	CT-HRP	*p*-Value		CT-HRP	*p*-Value
	Odds Ratio (95% C.I.)			Odds Ratio (95% C.I.)	
Gender (Men %)	4.322 (1.236–15.109)	0.0219	Retinopathy (%)	2.400 (0.925–6.224)	0.0718
Age (years old)	1.022 (0.976–1.070)	0.3603	Nephropathy (%)	0.911 (0.338–2.452)	0.8532
Duration of diabetes (years)	1.011 (0.964–1.062)	0.6475	Neuropathy (%)	2.829 (1.153–6.989)	0.0232
BMI (kg/m^2^)	0.941 (0.845–1.048)	0.2662	Medical treatment		
HT (%)	0.516 (0.205–1.311)	0.1651			
DL (%)	1.544 (0.608–3.916)	0.3608	Insulin treatment (%)	1.582 (0.683–3.663)	0.2841
Current smoking (%)	4.333 (1.832–10.250)	0.0008	Metformin (%)	0.683 (0.292–1.601)	0.3805
T-Cho (md/dL)	1.013 (1.001–1.024)	0.0293	Pioglitazone (%)	0.992 (0.205–4.806)	0.9921
TG (md/dL)	1.009 (1.004–1.014)	0.0008	DPP-4I (%)	0.696 (0.292–1.664)	0.4154
LDL-C (md/dL)	1.005 (0.992–1.019)	0.4209	SGLT2I (%)	0.911 (0.338–2.452)	0.8532
HDL-C (md/dL)	0.964 (0.925–1.005)	0.0852	ACE-I/ARB (%)	0.498 (0.176–1.405)	0.1876
non-HDL (md/dL)	1.017 (1.005–1.028)	0.0058	Statin (%)	0.957 (0.418–2.188)	0.9165
Lp(a) (mg/dL)	0.960 (0.919–1.003)	0.0703	Non-invasive examination		
UA (mg/dL)	1.233 (0.879–1.729)	0.2255			
HbA1c (%)	1.269 (0.983–1.638)	0.0673	Agatston score	1.000 (0.999–1.001)	0.9365
Fasting glucose (g/dL)	1.017 (1.005–1.028)	0.0037	Agatston score category		
GA (mg/dL)	1.077 (1.004–1.157)	0.0398	0–99 (reference)	1 (1)	N.A.
hs-CRP (mg/dL)	1.004 (0.359–2.807)	0.9934	100– (%)	1.187 (0.435–3.237)	0.7379
Cre (mg/dL)	1.021 (0.593–1.757)	0.9401	Visceral fat-type obesity (%)	2.019 (0.566–7.205)	0.2792
eGFR (mL/min/1.73 m^2^)	0.982 (0.958–1.007)	0.1535	Mean IMT (mm)	1.341 (0.268–6.700)	0.7210
			Max IMT (mm)	0.615 (0.252–1.501)	0.2854
			Increased IMT (%)	0.767 (0.301–1.953)	0.5775
			Stress ECG positive (%)	0.590 (0.128–2.730)	0.4997

CT-HRP, computed tomography-verified high-risk plaque; 95% C.I., 95% confidence interval; BMI, body mass index; HT, hypertension; DL, dyslipidemia; T-Cho, total cholesterol; TG, triglyceride; LDL-C, low-density lipoprotein cholesterol; HDL-C, high-density lipoprotein cholesterol; non-HDL, non-high-density lipoprotein; LP, lipoprotein; UA, uric acid; HbA1c, hemoglobin A1c; GA, glycated albumin; hs-CRP, high-sensitivity C-reactive protein; Cre, creatinine; eGFR, estimated glomerular filtration rate, DPP-4I, dipeptidyl peptidase-4 inhibitor; SGLT2I, sodium glucose cotransporter-2 inhibitor; ACE-I, angiotensin-converting enzyme inhibitor; ARB, angiotensin II receptor blocker; IMT, intima-media thickness; ECG, electrocardiogram.

**Table 5 jcm-11-01226-t005:** Multivariate analysis using logistic regression to identify the independent predictor of CT-HRP.

	CT-HRP	*p*-Value
	Odds Ratio (95% C.I.)	
Gender (Male %)	2.060 (0.465–9.122)	0.3412
Current smoking (%)	5.013 (1.683–14.931)	0.0038
T-Cho (md/dL)	0.992 (0.941–1.046)	0.7658
TG (md/dL)	1.006 (0.999–1.001)	0.1063
non-HDL (md/dL)	1.019 (0.963–1.077)	0.5178
Fasting glucose (g/dL)	1.012 (0.996–1.028)	0.1387
GA (mg/dL)	1.037 (0.931–1.155)	0.5077
Neuropathy (%)	2.466 (0.790–7.699)	0.1203

CT-HRP, computed tomography-verified high-risk plaque; 95% C.I., 95% confidence interval; T-Cho, total cholesterol; TG, triglyceride; non-HDL, non-high-density lipoprotein; GA, glycated albumin.

**Table 6 jcm-11-01226-t006:** Univariate analysis to identify the independent predictor of optimal revascularization within 90 days.

	Optimal Revascularization within 90 Days	*p*-Value		Optimal Revascularization within 90 Days	*p*-Value
	Odds Ratio (95% C.I.)			Odds Ratio (95% C.I.)	
Gender (Male %)	0.893 (0.256–3.153)	0.8594	Retinopathy (%)	3.913 (1.143–13.397)	0.0298
Age (years old)	1.038 (0.9691–1.1113)	0.2883	Nephropathy (%)	0.628 (0.131–2.999)	0.5596
Duration of diabetes (years)	1.042 (0.977–1.111)	0.2120	Neuropathy (%)	3.087 (0.912–10.445)	0.0700
BMI (kg/m^2^)	0.951 (0.817–1.107)	0.5176	Medical treatment		
HT (%)	0.523 (0.136–2.012)	0.3458			
DL (%)	1.561 (0.405–6.021)	0.5177	Insulin treatment (%)	6.313 (1.635–24.377)	0.0075
Current smoking (%)	2.921 (0.889–9.602)	0.0775	Metformin (%)	0.874 (0.265–2.880)	0.8251
T-Cho (md/dL)	1.017 (1.001–1.033)	0.0332	Pioglitazone (%)	0.000 (0.000–0.000)	0.9946
TG (md/dL)	0.999 (0.992–1.006)	0.6683	DPP-4I (%)	0.721 (0.208–2.502)	0.6016
LDL-C (md/dL)	1.027 (1.007–1.048)	0.0077	SGLT2I (%)	0.628 (0.131–2.999)	0.5596
HDL-C (md/dL)	0.996 (0.948–1.046)	0.8639	ACE-I/ARB (%)	0.462 (0.097–2.195)	0.3316
non-HDL (md/dL)	1.018 (1.002–1.034)	0.0263	Statin (%)	3.386 (0.882–13.004)	0.0757
Lp(a) (mg/dL)	1.001 (0.964–1.039)	0.9676	Non-invasive examination		
UA (mg/dL)	1.003 (0.609–1.652)	0.9908			
HbA1c (%)	1.027 (0.705–1.496)	0.8901	Agatston score	1.0006 (0.9998–1.0014)	0.1649
Fasting glucose (g/dL)	1.003 (0.994–1.012)	0.4847	Agatston score category		
GA (mg/dL)	1.083 (0.985–1.191)	0.0975	0–99 (reference)	1 (1)	N.A.
hs-CRP (mg/dL)	0.593 (0.021–15.987)	0.7471	100– (%)	3.228 (0.952–10.943)	0.0600
Cre (mg/dL)	1.001 (0.444–2.258)	0.9977	Visceral fat-type obesity (%)	2.658 (0.330–21.427)	0.3585
eGFR (mL/min/1.73 m^2^)	0.968 (0.931–1.006)	0.0970	Mean IMT (mm)	1.151 (0.191–12.062)	0.9068
			Max IMT (mm)	1.746 (0.897–3.398)	0.1009
			Increased IMT (%)	3.567 (1.072–11.864)	0.0381
			Stress ECG positive (%)	2.978 (0.726–12.214)	0.1297

95% C.I., 95% confidence interval; BMI, body mass index; HT, hypertension; DL, dyslipidemia; T-Cho, total cholesterol; TG, triglyceride; LDL-C, low-density lipoprotein cholesterol; HDL-C, high-density lipoprotein cholesterol; non-HDL, non-high-density lipoprotein; LP, lipoprotein; UA, uric acid; HbA1c, hemoglobin A1c; GA, glycated albumin; hs-CRP, high-sensitivity C-reactive protein; Cre, creatinine; eGFR, estimated glomerular filtration rate, DPP-4I, dipeptidyl peptidase-4 inhibitor; SGLT2I, sodium glucose cotransporter-2 inhibitor; ACE-I, angiotensin-converting enzyme inhibitor; ARB, angiotensin II receptor blocker; IMT, intima-media thickness; ECG, electrocardiogram.

**Table 7 jcm-11-01226-t007:** Multivariable analysis to identify the independent predictor of optimal revascularization within 90 days.

	Optimal Revascularization within 90 Days	*p*-Value
	Odds Ratio (95% C.I.)	
T-Cho (md/dL)	0.962 (0.891–1.040)	0.3298
LDL-C (md/dL)	1.058 (0.987–1.134)	0.1114
non-HDL (md/dL)	1.009 (0.936–1.089)	0.8097
Retinopathy (%)	3.083 (0.755–12.592)	0.1168
Medical treatment		
Insulin treatment (%)	5.677 (1.223–26.349)	0.0266
Non-invasive examination		
Increased IMT (%)	2.588 (0.673–9.951)	0.1663

95% C.I., 95% confidence interval; T-Cho, total cholesterol; LDL-C, low-density lipoprotein cholesterol; non-HDL, non-high-density lipoprotein; IMT, intima-media thickness.

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
