# Peer review of "Identification of Risk Factors for Coronary Artery Disease in Asymptomatic Patients with Type 2 Diabetes Mellitus"

_jcm, 2022, doi:10.3390/jcm11051226_

Round 1

Reviewer 1 Report

In this study Takamura,et al. aimed to identify high-risk groups in asymptomatic patients with type 2 diabetes mellitus. The results show that an Agatston score ≥100 and current smoking are independent predictive factors for obstructive CAD. Furthermore, the insulin treatment represents the only predictive factor for optimal revascularization within 90 days. So, non-invasive examinations except for Agatston score are not independent predictors for CAD in asymptomatic type 2 diabetic patients.

We thank the authors for this interesting study. The study design is fitting and the statistical analysis is rigorous. This reviewer has some minor concerns

  1. The patients enrolled in this study seems to be at low risk for coronary artery disease. May the characteristics of the population have reduced the power of detecting predictors of coronary artery disease?
  2. The sample size seems to be insufficient to detect clinical predictors of coronary artery disease among asymptomatic patients. Please discuss.
  3. There are several typos. Try to check all of them (for example in Table 1 “Agatston socre” and “Agatston socore”).

Author Response

Response to reviewer 1 suggestion.

#1. The patients enrolled in this study seems to be at low risk for coronary artery disease. May the characteristics of the population have reduced the power of detecting predictors of coronary artery disease?

To reviewer 1

Thank you for your suggestion.

According to the ESC guidelines, diabetic patients have a moderate or greater risk of coronary artery disease. In the FACTOR 64 and DIAD study, the incidence of coronary artery disease in asymptomatic patients with DM was low and similar to the results of this study. However, the small population and low prevalence may have contributed to the low power of the predictor, as you pointed out, and we have added to the section of Limitation.

#2. The sample size seems to be insufficient to detect clinical predictors of coronary artery disease among asymptomatic patients. Please discuss.

To reviewer 1

Thank you for your opinion.

As you pointed out, due to the small sample size, the total number of CAD patients and patients underwent revascularization were also small, which may have affected the detection of predictors. We have added it to Limitation including the contents of #1.

#3. There are several typos. Try to check all of them (for example in Table 1 “Agatston socre” and “Agatston socore”).

To reviewer 1

Thank you for your kind advice.

We re-read our article and corrected spelling mistake. Finally, we had our article proofread by a translation agency (Medical English Service).

Reviewer 2 Report

This manuscript is interesting. However, this reviewer raises some issues that the authors have to address.

1- The main limitation of the study is its observational prospective design which prevents the definition of a cause/effect relationship between parameters/risk factors and CAD. This issue needs to be addressed by the authors and added to the limitations of the study.

2- In Conclusion the Authors state “Asymptomatic patients with type 2 DM were a heterogeneous group that included 434 those at a low risk of CAD”. The 2019 ESC/EASD Guidelines defined diabetic subjects with moderate, high or very high CV risk. Therefore, defining people with diabetes at low CV risk is wrong.

3- Albuminuria is missing among the risk factors for CAD analyzed by authors. A previous study with CCTA showed that albuminuria is associated with a high prevalence of CAD in asymptomatic type 2 diabetic patients (Diab Vasc Dis Res. 2012 Jan;9(1):10-7. doi: 10.1177/1479164111426439.). Add this issue to the discussion and limitations of the study.

4- This prospective study analyzed risk factors for CAD in asymptomatic diabetic subjects. The purpose of this type of study is to identify risk factors whose treatment should lead to a reduction in CV risk. In fact, a few months ago a multicenter randomized controlled trial showed that an intensified multifactorial intervention that reduces the main CV risk factors leads to a reduction in MACEs and mortality in type 2 diabetics in primary CV prevention (Cardiovasc Diabetol. 2021 Jul 16;20(1):145. doi: 10.1186/s12933-021-01343-1.). This important issue should be addressed in the discussion.

5- In all manuscript tables, UA is defined as urinary acid. Actually, I think UA indicates uric acid. If so, please correct.

6- Tables should be formatted more homogeneously.

Author Response

Response to reviewer 2 suggestion.

#1. The main limitation of the study is its observational prospective design which prevents the definition of a cause/effect relationship between parameters/risk factors and CAD. This issue needs to be addressed by the authors and added to the limitations of the study.

To reviewer 2

Thank you for your suggestion.                               

Although the present study was an observational study, CAD findings by CCTA during the hospitalization period were used as the outcome, which makes it difficult to interpret the causal relationship between CAD and risk factors, as you pointed out.  This is one of the problems of this study, and we have described it in the section of Limitation.

#2. In Conclusion the Authors state “Asymptomatic patients with type 2 DM were a heterogeneous group that included 434 those at a low risk of CAD”. The 2019 ESC/EASD Guidelines defined diabetic subjects with moderate, high or very high CV risk. Therefore, defining people with diabetes at low CV risk is wrong.

To reviewer 2        

Thank you for your advice.                                             

There were asymptomatic patients with DM without coronary artery disease in this study, but as you pointed out, patients with DM are also considered CV risk in the guidelines, and this is a misleading expression, so we have deleted and corrected it.

#3. Albuminuria is missing among the risk factors for CAD analyzed by authors. A previous study with CCTA showed that albuminuria is associated with a high prevalence of CAD in asymptomatic type 2 diabetic patients (Diab Vasc Dis Res. 2012 Jan;9(1):10-7. doi: 10.1177/1479164111426439.). Add this issue to the discussion and limitations of the study.

To reviewer 2                                                    

Thank you for your opinion.                                             

As you pointed out, albuminuria is an important risk factor for CAD in asymptomatic type 2 diabetic patients. In this study, we considered additional analysis of albuminuria, but were unable to do so due to missing data. This may have affected the predictive factors; it has been added to Limitation.

#4. This prospective study analyzed risk factors for CAD in asymptomatic diabetic subjects. The purpose of this type of study is to identify risk factors whose treatment should lead to a reduction in CV risk. In fact, a few months ago a multicenter randomized controlled trial showed that an intensified multifactorial intervention that reduces the main CV risk factors leads to a reduction in MACEs and mortality in type 2 diabetics in primary CV prevention (Cardiovasc Diabetol. 2021 Jul 16;20(1):145. doi: 10.1186/s12933-021-01343-1.). This important issue should be addressed in the discussion.

To reviewer 2                                                     

Thank you for your opinion.                                              

As you pointed out, it is very important to report that multifaceted therapeutic intervention reduced MACE and mortality in patients with high-risk DKD. It may be possible to improve the prognosis of asymptomatic diabetic patients by focusing on the coronary high-risk factors identified in our study. I have included the report you pointed out in the discussion.

#5. In all manuscript tables, UA is defined as urinary acid. Actually, I think UA indicates uric acid. If so, please correct.

To reviewer 2                                                     

Thank you for your advice.                                            

There were spelling mistakes as pointed out. The manuscript was reviewed, checked by a native English proofreader, and corrected.

#6. Tables should be formatted more homogeneously.                     

To reviewer 2                                                      

Thank you for your comments. I've unified the format of the Table as you suggested.

Round 2

Reviewer 2 Report

Authors addressed all issues raised by this reviewer.

However, please note that the authors of all references cited in the manuscript are named with the first name, while the surname and any middle name are abbreviated with the initial only. This is contrary to the editorial rules of JCM and most Journals. Authors need to correct this mistake.